# Naringenin and Quercetin Exert Contradictory Cytoprotective and Cytotoxic Effects on Tamoxifen-Induced Apoptosis in HepG2 Cells

**DOI:** 10.3390/nu14245394

**Published:** 2022-12-19

**Authors:** Zhixiang Xu, Yue Jia, Jun Liu, Xiaomin Ren, Xiaoxia Yang, Xueshan Xia, Xuejun Pan

**Affiliations:** 1Faculty of Environmental Science & Engineering, Kunming University of Science and Technology, Kunming 650500, China; 2Faculty of Life Science & Technology, Kunming University of Science and Technology, Kunming 650500, China

**Keywords:** hepatotoxicity, double dose-response, tamoxifen, naringenin, quercetin, apoptosis, reactive oxygen species

## Abstract

Tamoxifen is commonly used to treat estrogen receptor-positive breast cancer and hepatocellular carcinoma. Phytoconstituents are considered candidates for chemopreventive drugs in cancer treatment. However, it remains unknown what would happen if tamoxifen and phytoconstituents were administrated simultaneously. We aimed to observe the synergistic antitumor effects of tamoxifen and naringenin/quercetin on human hepatic carcinoma and to explore the potential underlying molecular mechanisms. The HepG2 cell line was used as an in vitro model. Cell proliferation, invasion, migration, cycle progression and apoptosis were investigated along with reactive oxygen species (ROS) production and mitochondrial membrane potential (ΔΨm) repression. The signaling pathways involved were identified using real-time quantitative polymerase chain reaction analysis. As the results show, tamoxifen in combination with higher concentrations of naringenin or quercetin significantly inhibited cell growth compared to either agent alone. These antiproliferative effects were accompanied by the inhibition of cell migration and invasion but the stimulation of cell apoptosis and loss of ΔΨm, which depended on the ROS-regulated p53 signaling cascades. Conversely, lower concentrations of naringenin and quercetin inhibited the tamoxifen-induced cell antiproliferative effects by regulating cell migration, invasion, cycle and apoptosis. Taken together, our findings revealed that phytoconstituents exerted contradictory cytoprotective and cytotoxic effects induced by tamoxifen in human hepatic cancer.

## 1. Introduction

Epidemiological studies have identified a positive correlation between the increased consumption of natural products which contained various phytoconstituents with a decreased risk of cancer [1]. Naringenin (flavanone) and quercetin (flavonol), two representative phytoconstituents widely used in the human diet, have been shown to be positively associated with beneficial health effects [2]. Such phytoconstituents are gradually becoming considered as chemopreventive candidates in cancer treatment [3,4]. Primary liver cancer is the sixth commonly diagnosed cancer worldwide and is estimated to be the fourth leading cause of cancer death, with approximately 841,000 new cases and 782,000 deaths annually [5]. Hepatocellular carcinoma (HCC) makes up the majority of cases, contributing to approximately 75–85% of all cancers [5]. Surgery is considered a good treatment option for HCC, but only a small minority of patients can undergo radical resection after diagnosis [6]. In this way, the early prevention of HCC seems particularly important.

Currently, hormone therapy and chemoprevention are two of the most well-studied approaches in cancer treatment, including HCC [7]. Several clinical and experimental studies have indicated that tamoxifen, as an antiestrogen, seems to be effective in the palliative treatment of HCC, the same as breast cancer, via estrogen receptor (ER)-dependent or independent pathways [6,8,9]. Tamoxifen therapy is not fully effective due to undesirable effects relative to ER-mediated signaling [10,11]. Specifically, tamoxifen does not bind to variant estrogen receptors, so it is only effective in a specific subset of HCC patients [12,13], thereby limiting its use in many patients. As tamoxifen is not always effective for all HCC cases, it would be better to find novel and effective endocrine drugs to be used in HCC treatment.

Previously, we reported that naringenin at higher concentrations can induce cell apoptosis and inhibit cell proliferation in breast carcinoma cells [14]. Importantly, tamoxifen combined with naringenin additively promoted cell apoptosis in MCF-7 cells [14]. Numerous studies have reported the antioxidant and anticancer effects of tamoxifen, naringenin and quercetin [15,16,17,18,19], but the combined effects of tamoxifen and flavonoids and the underlying cellular and molecular mechanisms are not clear in HCC cells. Of all the in vitro models of HCC, the HepG2 cell lines, with their immortal features and high proliferation potential [20], are metabolically competent and able to metabolize a variety of toxicants [21]. Meanwhile, HepG2 cells have a high content of organelles and mitochondrial DNA [22,23], making them suitable for mitochondrial-related studies.

Based on the aforementioned, we selected HepG2 cells as an HCC in vitro model to investigate the effects of tamoxifen combined with two phytoconstituents (i.e., naringenin and quercetin) concerning cell proliferation and apoptosis. Various endpoints of cytotoxic treatment, including cell viability, invasion, migration, cell cycle, cell apoptosis, reactive oxygen species (ROS) production and mitochondrial membrane potential (ΔΨm), were examined. In addition, the involvement of some signaling pathways was also explored. The data obtained could provide a systematic cognition of the combined effects of phytoconstituents and pharmaceutical antiestrogens on HCC treatment.

## 2. Materials and Methods

### 2.1. Biological and Chemical Materials

Cell cultural reagents, including fetal bovine serum (FBS), charcoal-stripped FBS (CS-FBS), phenol red- and/or free DMEM, phosphate buffered saline (PBS) and penicillin-streptomycin, were procured from Thermo Fisher Scientific (Shanghai, China). Tamoxifen, naringenin and quercetin standards were purchased from Sigma-Aldrich (St. Louis, MO, USA). Stock solutions at 100 mM were prepared in dimethyl sulfoxide (DMSO) and stored at −20 °C, and they were diluted as needed with an experimental medium (phenol red-free DMEM containing 5% CS-FBS) before use.

### 2.2. Cell Culture and Chemical Treatment

Human hepatic adenocarcinoma HepG2 cells were obtained from the American Type Culture Collection (ATCC, Manassas, VA, USA) and cultured as previously described [20]. Exponentially growing cells were seeded in 96-well microplates (5 × 10^3^/well) for cell viability assays, 12-well microplates (3 × 10^5^/well) for cell migration, 24-well transwell chambers (1 × 10^5^/well) for cell invasion and 6-well plates (3 × 10^5^/well) for cell cycle, apoptosis, oxidative stress (i.e., LDH, GSH and caspase activity), ΔΨm, ROS and gene expression analysis, respectively. For all endpoints, cells were treated with increasing concentrations of tamoxifen (0.01–40 μM), naringenin (0.1–500 μM) and quercetin (0.1–500 μM) or its vehicle (0.1% DMSO) for a specific time.

### 2.3. Cell Proliferation and Cytotoxicity Evaluation

The cell counting kit-8 (CCK-8) assay was performed to evaluate the cell proliferation and cytotoxicity elicited by tamoxifen and naringenin/quercetin alone or in combination at 24, 48 and 72 h using the SpectraMax M5 microplate system (Molecular Devices, San Jose, CA, USA), as previously described [14]. The corresponding IC50 values, defined as the concentration of the tested compounds that resulted in 50% growth inhibition, were calculated from the dose-response curves. The real-time cell impedance analyzer (RTCA; xCELLigence RTCA S16, ACEA Biosciences, San Diego, CA, USA) was also employed to continuously monitor the proliferative or antiproliferative effects of tamoxifen, naringenin and quercetin alone or in combination, as previously described [14,24].

### 2.4. Wound Healing Assay

A wound healing assay was adopted to analyze the effects of tamoxifen, naringenin and quercetin on cell motility, as previously described [14,25], with minor modifications. Following treatment with naringenin (10 and 200 μM), quercetin (10 and 100 μM) and tamoxifen (20 μM) alone or in combination based on the cell viability results, the wound closure was photographed at 100× magnification using the inverted fluorescence microscope (IX73, Olympus, Tokyo, Japan) at 0, 12, 24 and 48 h. Then, the photographs were analyzed with Image-Pro Plus software (Media Cybernetics, Silver Spring, CA, USA).

### 2.5. Transwell Assay

The effects of tamoxifen, naringenin and quercetin on cell invasion were investigated using the transwell assay [14]. Following incubation with objective compounds for 24 h, cells were cultured with MTT reagents, and the optical density at 570 nm was detected with a microplate system (SpectraMax M5) to indirectly reflect cell invasiveness.

### 2.6. Cell Cycle and Apoptosis Assessment Using Flow Cytometry

Cell cycle and apoptosis were detected using a cell cycle and apoptosis analysis kit (Beyotime, China) and Annexin V-FITC apoptosis detection kit (Nanjing KeyGen Biotech, Nanjing, China), respectively, with a Guava easyCyte 6HT-2L flow cytometry system (Merck Millipore, Boston, MA, USA), following treatment with objective compounds for 24 h, as previously described [14]. The cell cycle distribution (G0/G1, S and G2/M phases) and apoptosis proportion (early apoptosis and late apoptosis) were then calculated using NovoExpress software (ACEA Biosciences, San Diego, CA, USA). The measurement of cells in the sub-G1 phase with PI staining was also calculated to estimate the cellular apoptotic induction by tamoxifen, naringenin and quercetin.

### 2.7. Fluorescence Microscopy

Cell apoptosis was identified as irregular heterogeneous patchy inclusions in the nucleus due to contrasting chromatin condensation with the nonapoptotic cells which were uniformly dispersed and homogenously stained [26]. After being exposed to objective compounds for 24 h, cells were stained with a 4,6-diamidino-2-phenylindole (DAPI) fluorescent probe (Beyotime, China). Then, the strained cells were scrutinized for nuclear morphology at 400× magnifications using an Olympus inverted fluorescence microscope (IX73).

### 2.8. Determination of LDH Leakage, GSH Content and Caspase Activity

The LDH release, GSH content, caspase-3 activity and caspase-9 activity were determined using the LDH cytotoxicity kit, glutathione reductase assay kit, caspase-3 activity assay kit and caspase-9 activity assay kit (Nanjing JianCheng Bio Institute, Nanjing, China), respectively, according to the manufacturer protocol [18,27,28].

### 2.9. ROS Determination

Intracellular ROS was determined with a 2,7-dichlorodi-hydrofluorescein diacetate (DCFH-DA) probe (Beyotime) using the Guava easyCyte 6HT-2L flow cytometry system (Merck Millipore), following treatment with objective compounds for 24 h, according to our previous study [14].

### 2.10. ΔΨm Determination

Mitochondrial membrane potential loss (ΔΨm) was measured with a JC-1 probe (Nanjing KeyGen Biotech, Nanjing, China). Briefly, following treatment with objective compounds for 24 h, cells were stained with a JC-1 probe (10 μM) at 37 °C for 20 min, and the fluorescence was then measured with a flow cytometry system (Merck Millipore, Boston, MA, USA) and inversed fluorescence microscope (IX73, Olympus, Tokyo, Japan).

### 2.11. RNA Extraction and RT-qPCR

The expression levels of genes involved in the process of cell proliferation and apoptosis were detected by quantitative polymerase chain reaction (qPCR) using the StepOnePlus real-time PCR system (RT-qPCR; Applied Biosystems, Waltham, CA, USA), according to our previous studies [14,20]. The validated primers of desired genes (i.e., MMP-9, MMP-2, E-cadherin, N-cadherin, cyclin D1, cyclin E, p53, p21, Bcl-2, Bax and β-actin) are listed in Appendix A. The mRNA levels of desired genes were calculated using the 2^−ΔΔCt^ method, and β-actin was used as an endogenous control for internal normalization [14,29].

### 2.12. Statistical Analysis

All experiments were independently performed in triplicate. Data were analyzed using SPSS 17.0 software (Chicago, IL, USA), and results were expressed as mean ± standard deviation (SD). Statistical significance was assessed by one-way analysis of variance (ANOVA), followed by a post hoc LSD test. *p* < 0.05 was considered statistically significant.

## 3. Results

### 3.1. Tamoxifen, Naringenin and Quercetin Induced Morphological Changes and Regulated Cell Viability

The proliferative and antiproliferative effects of tamoxifen, naringenin and quercetin were examined using CCK-8 and RTCA assays, as presented in Figure 1 and Figure 2, respectively. The CCK-8 assays showed that the incubation of HepG2 cells with naringenin and quercetin (0.1 to 10 μM) and with tamoxifen (0.01 to 0.5 μM) slightly increased the cell viability, whereas they significantly inhibited cell proliferation in time- and dose-dependent manners at higher concentrations (Figure 1A–C). Similar results were found in other studies [30]. In particular, naringenin showed less low toxicity to HepG2 cells compared with quercetin. Appendix A presents the half-maximal inhibitory concentrations (IC50) of tamoxifen, naringenin and quercetin at 24, 48 and 72 h. These results indicate that lower concentrations of both the phytoconstituents (naringenin and quercetin) and pharmaceutical antiestrogen (tamoxifen) treatment can increase the cell survival rates in HepG2 cells, but higher concentrations of them exert cytotoxicity.

Considering that naringenin and quercetin at higher concentrations (200 and 100 μM, respectively) and tamoxifen at 20 μM markedly decreased cell viability and that naringenin and quercetin at lower concentrations (10 μM) slightly promoted cell proliferation, we selected these optimum concentrations for further studies. Figure 1D shows that the coexposure of tamoxifen with high concentrations of naringenin or quercetin synergistically inhibited cell viability compared with the single treatment, whereas these tamoxifen-induced antiproliferative effects were partly compensated by lower concentrations of naringenin or quercetin.

The photograph showed that treating cells with higher concentration objectives drastically altered the round morphological shape of the cells, but no obvious changes were found when the cells were exposed to lower concentrations of them (Appendix A). The NCI values reflected the state of cell growth and death [31]. As described in Figure 2, tamoxifen, naringenin and quercetin alone or in combination regulated cell proliferation similar to the CCK8 results. In addition, the RTCA assay showed a much higher sensitivity and high-speed response capacity than that of the CCK-8 assay.

### 3.2. Tamoxifen, Naringenin and Quercetin Regulated Cell Migration and Invasion

As shown in Appendix A and Figure 3A,B, lower concentrations of naringenin and quercetin slightly increased cell migration and invasion but were not statistically significant (*p* > 0.05). In contrast, higher concentrations of them significantly inhibited HepG2 migration in a time-dependent manner. Similarly, higher concentrations of naringenin and quercetin also repressed cell invasion. For the cotreatment group, tamoxifen combined with higher concentrations of naringenin or quercetin significantly inhibited cell migration and invasion, whereas the tamoxifen-induced effects were partially compensated by low concentrations of naringenin and quercetin. In addition, no significant differences were found when cells were cotreated with naringenin and quercetin compared with the single-treated group.

Corresponding to the wound healing and transwell invasion results, tamoxifen and higher concentrations of the naringenin and quercetin treatment alone or in combination significantly down-regulated the mRNA transcription of MMP-2, MMP-9 and N-cadherin compared with the control group (Figure 3C,D). However, lower concentrations of naringenin and quercetin increased the mRNA transcription of MMP-9, MMP-2 and N-cadherin that were inhibited by tamoxifen. The mRNA transcription of E-cadherin was almost exactly the opposite to that of N-cadherin.

### 3.3. Tamoxifen, Naringenin and Quercetin Regulated Cell Cycle Progression

Cell cycle dysregulation has been affirmed to contribute to aberrant cell proliferation and cancer development [32]. As shown in Appendix A and Figure 4A–C, the cell cycle in G0/G1 phase was increased in HepG2 cells treated with tamoxifen compared with the control group (51.93 ± 0.46% vs. 57.26 ± 0.42%). As for naringenin and quercetin, they exhibited different behaviors in the regulation of the cell cycle at different exposure concentrations. In detail, naringenin at higher concentrations induced cell cycle arrest in the G0/G1 and S phases, whereas higher concentrations of quercetin inhibited cell cycles in the S stage. Importantly, tamoxifen combined with higher concentrations of naringenin and quercetin further induced cell arrest in the G0/G1 phase and S phase, respectively. However, lower concentrations of naringenin and quercetin could promote the G1 phase to S phase transition.

The mRNA transcription levels of two cell cycle-related genes, cyclin D1 and cyclin E, were then examined as they are involved in accelerating cell growth [33]. Figure 4D shows that the mRNA transcription levels of cyclin D1 and cyclin E were significantly decreased following treatment with tamoxifen alone or in combination with higher concentrations of naringenin and quercetin. However, lower concentrations of naringenin and quercetin increased the mRNA transcription of both cyclin D1 and cyclin E. Taken together, cell cycle arrest induced by tamoxifen, naringenin and quercetin is positively correlated with the activity inhibition of the G0/G1 and S phase-related genes in HepG2 cells.

### 3.4. Tamoxifen, Naringenin and Quercetin Regulated Cell Apoptosis

Apoptosis was performed concurrently to determine whether the growth was associated with apoptotic regulation. The flow cytometry analyses revealed enhanced apoptosis following treatment with tamoxifen (20 μM), naringenin (200 μM) or quercetin (100 μM), which was based on the formation of massive cellular aggregates with Annexin V-FITC positive cells (Appendix A and Figure 5A–C) and sub-G1 DNA content (Appendix A and Figure 5D). From Figure 5A–C, the number of apoptotic cells increased after treatment with tamoxifen and high concentrations of naringenin and quercetin. Moreover, numerous cells were in the apoptotic state, and the number of necrotic cells slightly increased (Appendix A). The percentages of total cell apoptosis increased from 3.50 ± 0.06% for the control group to 6.96 ± 0.18%, 9.47 ± 0.33%, 11.57 ± 0.42%, 21.21 ± 0.20% and 36.94 ± 0.24%, following exposure to tamoxifen (20 μM), naringenin (200 μM), quercetin (100 μM) and their combination, respectively, for 24 h. In contrast, the cell apoptosis ratios decreased in the presence of lower concentrations of naringenin or quercetin (10 μM).

DAPI staining also revealed the different degrees of cell shrinkage, nuclear condensation or fragmentation after the cells were exposed to tamoxifen, naringenin and quercetin (Figure 6). Specifically, tamoxifen-induced apoptosis or damage was attenuated in the presence of 10 μM naringenin or quercetin but was increased in the presence of naringenin (200 μM) or quercetin (100 μM). Therefore, the cytotoxic effects of tamoxifen or higher concentrations of naringenin and quercetin on HepG2 cells were closely associated with the induction of cell apoptosis and even necrosis. Particularly, higher concentrations of naringenin and quercetin combined with tamoxifen synergistically induced cell apoptosis, but lower concentrations of them suppressed the tamoxifen-induced apoptosis. These results are in accordance with the proliferative or apoptotic effects detected by the CCK-8 and RTCA assays.

Four common apoptosis-regulated genes, including p21, p53, Bcl-2 and Bax, were further assessed using RT-qPCR. Bcl-2 is extensively considered a therapeutic target for apoptosis-inducing anticancer approaches as it is often overexpressed in many cancers [1,34]. In contrast, p21, p53 and Bax are considered the apoptosis-induced genes [35]. As shown in Figure 7, the Bcl-2 expression was downregulated, but the expression of p21, p53 and Bax were upregulated after cells were treated with tamoxifen alone or combined with higher concentrations of naringenin and quercetin. In contrast, lower concentrations of naringenin and quercetin treatment decreased the mRNA transcription of the p21 and p53 genes to some extent. Moreover, no significance was observed in the expression of Bax, but the Bcl-2 and Bcl-2/Bax ratios were increased. Taken together, cell growth arrest and death largely depend on the apoptotic activity of higher concentrations of tamoxifen, naringenin and quercetin.

### 3.5. Tamoxifen, Naringenin and Quercetin Regulated the Leakage of LDH and Glutathione Content

LDH leakage was investigated as another indicator of cell toxicology [27]. Herein, HepG2 cells exposed to tamoxifen and higher concentrations of naringenin and quercetin caused a significant increase in the leakage of LDH, whereas lower concentrations of naringenin and quercetin did not observably change the release of LDH (Figure 8A). In addition, cotreatment cells with tamoxifen and higher concentrations of naringenin and quercetin significantly induced the leakage of LDH in an additive manner, whereas lower concentrations of naringenin and quercetin inhibited the tamoxifen-induced release of LDH. From Figure 8B, a significant negative linear correlation curve was observed between cell viability and LDH release (R^2^ = 0.9157) following treatment with different compounds.

High intracellular glutathione levels could oppose the apoptosis induced by xenobiotics [28]. Figure 8C shows that tamoxifen and higher concentrations of naringenin and quercetin treatment significantly decreased the GSH content in HepG2 cells, but no obvious changes were found when cells were exposed to lower concentrations of naringenin and quercetin. A positive linear correlation curve was also observed between cell viability and GSH content (R^2^ = 0.8568), as shown in Figure 8D. The above results demonstrate that the increase in mitochondrial membrane permeability induced by tamoxifen, naringenin, or quercetin is involved in cell death.

### 3.6. Tamoxifen, Naringenin and Quercetin Regulated the Activation of Caspases

Caspases are considered important mediators of cell apoptosis and contribute to the overall apoptotic morphology by cleaving various cellular substrates [36]. Caspase-3 is the major downstream effector of caspase in the apoptotic pathway [18], so we compared its activity with that of its upstream trigger, caspase-9, in treated and untreated control cells (Figure 8E,F). As shown in the present work, although lower concentrations of naringenin and quercetin did not observably change the activities of caspase-3 and caspase-9, they were significantly increased to 57.02%, 74.81% and 82.84% for caspase-3 and 42.89%, 60.82% and 71.87% for caspase-9, following treatment with tamoxifen (20 μM), naringenin (200 μM) and quercetin (100 μM), respectively. Importantly, the caspase-3 and caspase-9 activities were significantly increased after cells received combined treatment with tamoxifen and higher concentrations of naringenin and quercetin. In contrast, the tamoxifen-induced activation of caspases was eliminated with lower concentrations of naringenin and quercetin. These results indicate that apoptosis induced by tamoxifen and higher concentrations of naringenin and quercetin depends on intrinsic apoptosis pathways and that the mitochondrial amplification step is important in tamoxifen-, naringenin- and quercetin-induced cell apoptosis.

### 3.7. Tamoxifen, Naringenin and Quercetin Regulated ΔΨm in HepG2 Cells

The depolarization of mitochondrial membrane potential (ΔΨm) is one of the hallmark events of apoptosis [18]. As shown in Appendix A and Figure 9A,B, cells exposed to tamoxifen, naringenin and quercetin alone, especially at higher concentrations, triggered mitochondrial injury in HepG2 cells, and coadministration of tamoxifen and higher concentrations of naringenin and quercetin caused an increase in the loss of ΔΨm. In contrast, the ΔΨm loss induced by tamoxifen decreased more or less following treatment with lower concentrations of naringenin and quercetin.

### 3.8. Tamoxifen, Naringenin and Quercetin Regulated ROS Generation

Although ROS has been considered an inducer of cell death in several studies [18], it can also act as a growth stimulator to promote cell survival and proliferation by regulating signaling cascades [27]. Namely, a moderate increase in ROS generation is involved in cell proliferation, but excessive production of ROS would cause cytotoxicity and genotoxic effects on cells. As illustrated in Figure 9C,D, only 19.24% of the DCFH-DA positive cell populations were detected in the control group, whereas cells exposed to tamoxifen and higher concentrations of naringenin and quercetin either alone or in combination significantly increased the proportion of DCFH-DA positive cell populations. Interestingly, cells treated with 10 μM naringenin and quercetin and their combination induced ROS generation by 32.29%, 26.66% and 36.53%, respectively, suggesting a moderate increase in ROS generation by flavonoids is involved in cell proliferation. Similar phenomena have also been found in a previous study [27]. Moreover, tamoxifen combined with higher concentrations of naringenin and quercetin synergistically induced ROS generation, but low concentrations of naringenin and quercetin could eliminate the tamoxifen-induced ROS generation. In addition, naringenin and quercetin induced ROS generation in an additive manner at both lower and higher concentrations. These results suggest that tamoxifen-, naringenin- and quercetin-induced cell apoptosis is also associated with ROS accumulation.

## 4. Discussion

The present study shows for the first time that low concentrations of naringenin and quercetin can stimulate the proliferation of HepG2 cells and then inhibit their proliferation. In addition, they show contradictory cytoprotective and cytotoxic effects on tamoxifen-induced antiproliferation in human hepatic cancer HepG2 cells. Although the cell model used in the present study is different, these results are consistent with our previous studies on breast cancer cell lines [14]. To elucidate the conflicting roles of naringenin and quercetin on HepG2 growth, we investigated several specific molecular signaling pathways associated with cell proliferation and antiproliferation.

Consistent with these proliferative or inhibitory effects, high concentrations of tamoxifen, naringenin and quercetin alone or in combination significantly reduced adherent cell numbers, normal morphology and cell migration/invasion capacity compared with the individual treatment group, whereas lower concentrations of naringenin and quercetin had no significant influence on cell migration and invasion, only slightly promoting them. An increase in the number of dead or plasma membrane-damaged cells promotes LDH release in the culture supernatant [37]. Herein, higher concentrations of tamoxifen, naringenin and quercetin alone or in combination significantly increase LDH release, whereas lower concentrations of naringenin and quercetin can counteract the fractional effects induced by tamoxifen. Therefore, cell membrane damage may be involved in cell survival.

Cell cycle arrest and apoptosis are considered two important mechanisms involved in cancer therapy [17,32,38]. In the present study, higher concentrations of tamoxifen, naringenin and quercetin arrest cells in the G0/G1 or S phase, whereas lower concentrations of them promote the progression of cell cycles into the G2/M phase. p53 is redox-sensitive and regulates multiple cellular responses, such as cell survival and cell death, by blocking cell cycle progression in response to DNA damage [39]. Specifically, p53 activation contributes to the induction of cell cycle arrest in both the G1 and G2/M phases [35,40]. Cyclin D1 is considered a proto-oncogene for its ability to promote cell proliferation [15]. Herein, higher concentrations of naringenin and quercetin decrease the expression of cyclin D1 and cyclin E, therefore arresting cells in the G0/G1 and S phases. These results are consistent with the findings of previous studies [15,17,18]. Nonetheless, other studies reported naringenin and quercetin induced cell apoptosis by interrupting cell cycles in the G2/M phase [35]. These differences may be related to the different cell lines or specific culture conditions used in our experiments. In particular, the use of CS-FBS-supplemented medium could promote the transition of G1-S phase to some extent.

Likewise, the results obtained from both Annexin V-FITC/PI and DAPI straining assays show a significant increase in cell apoptosis induction when cells were treated with higher concentrations of tamoxifen, naringenin and quercetin alone or in combination. Similarly, they were previously identified to induce significant cytotoxicity in different cell types, such as MCF-7 [14,41], HepG2 [16,35] and human leukemic U937 cells [36]. Additionally, tamoxifen combined with higher concentrations of naringenin or quercetin synergistically induced HepG2 apoptosis. Instead, lower concentrations of naringenin and quercetin can inhibit tamoxifen-induced apoptosis.

Generally, apoptosis can be divided into an initiation stage and an executive stage. With respect to the initiation stage, cytochrome C is released from mitochondria to dimerize with apoptotic protease activating factor 1, thereby promoting apoptosis [23,36]. In the execution phase, caspase (i.e., caspase-9, caspase-8, caspase-7 and caspase-3) as the central components of apoptosis response are generated [23]. The caspase activity, therefore, was used as a marker for cell apoptosis [27]. In the present study, the activation of caspase-9 and caspase-3 was significantly increased after exposing HepG2 cells to high concentrations of tamoxifen, naringenin and quercetin. As suspected, the ΔΨm also markedly decreased. Based on our data, such apoptosis induction is associated with mitochondrial-mediated pathways.

Mitochondria-mediated apoptosis, of course, is regulated by the Bcl-2 protein family, which induces apoptosis by promoting the expression of the proapoptotic gene Bax or by inhibiting the expression of the antiapoptotic gene Bcl-2 [23,42]. A previous report indicated that Bcl-2 overexpression inhibited the induction of cell apoptosis in response to various chemical agents, including naringenin in several breast cancer cell lines [14]. The p53 gene, its downstream gene p21 and the p53/p21 complex have also been reported to upregulate Bax but downregulate Bcl-2 [23,43,44]. Herein, cells treated with higher concentrations of tamoxifen, naringenin and quercetin alone or in combination significantly increased the gene expression of p21, p53 and Bax but the opposite for Bcl-2, which are following the values reported in the literature [17,19,35,36]. In contrast, lower concentrations of naringenin and quercetin alone or combined with tamoxifen exposure resulted in an increase in Bcl-2 expression but decreases in p21, p53 and Bax expression. In addition, Bcl-2 overexpression or dropping rates of Bcl-2/Bax can block cytochrome C release from the mitochondria into the cytosol and then suppress the caspase activation of the downstream mitochondria to inhibit apoptosis [45]. For instance, naringenin-mediated apoptosis can be attenuated by Bcl-2 expression upregulation in U937 cells, but it was restored by a small molecule Bcl-2 inhibitor [36], which is in line with our present study in HepG2 cells.

ROS is essential for various biological processes in normal cells [28]. Specifically, human pathogenesis, including cancer, occurs when the redox balance is abnormal [1,46]. As mitochondria are sensitive targets of oxidative damage, impaired mitochondrial function is critical for ROS-triggered apoptotic pathways. Various in vivo or in vitro studies suggest that the toxic effects of exogenous biological products may be due to oxidative stress caused by the excessive production of reactive oxygen species [44,47]. Excess ROS results in the opening of the mitochondrial permeability transition pores, loss of MMPs and subsequent release of cytochrome C from mitochondria into the cytoplasm [23]. Previous reports have linked naringenin, quercetin and tamoxifen to ROS-mediated cell proliferation in several human carcinoma cells, such as breast cancer MCF-7 [14], epidermoid cancer A431 [18] and pancreatic cancer SNU-213 cells [47]. In fact, the antioxidant defense enzyme system plays a vital role in preventing oxygen-related damage and further preventing tumorigenesis [46]. Herein, obvious increases in intracellular ROS, LDH release and GSH depletion are observed following treatment with higher concentrations of tamoxifen, naringenin and quercetin. Clearly, a modest increase in ROS production can act as growth stimuli to regulate the signaling cascade [27], resulting in cell survival and proliferation. Similar results are also certified in our present study as lower concentrations of naringenin and quercetin inhibit cell apoptosis through a slight increase in ROS generation.

## 5. Conclusions

Collectively, naringenin and quercetin exhibit contradictory cytoprotective and cytotoxic effects on tamoxifen-induced antiproliferation in human hepatic cancer HepG2 cells. In particular, higher concentrations of both tamoxifen and phytoconstituent exposure stimulate ROS production and trigger mitochondria-mediated apoptosis. These apoptosis signaling pathways are involved in cell migration/invasion suppression, cell cycle arrest in the G0/G1 and S phases, loss of ΔΨm and the activation of caspases. Tamoxifen combined with higher concentrations of naringenin or quercetin synergistically induced cell apoptosis. Attractively, quercetin shows stronger antitumor activity than naringenin. In contrast, lower concentrations of naringenin or quercetin can partly compensate for tamoxifen-induced antiproliferative effects. Further research, of course, is needed to clarify the precise targets of the combination of phytoconstituents and tamoxifen in hepatic cancer cells.

## Figures and Tables

**Figure 1 nutrients-14-05394-f001:**
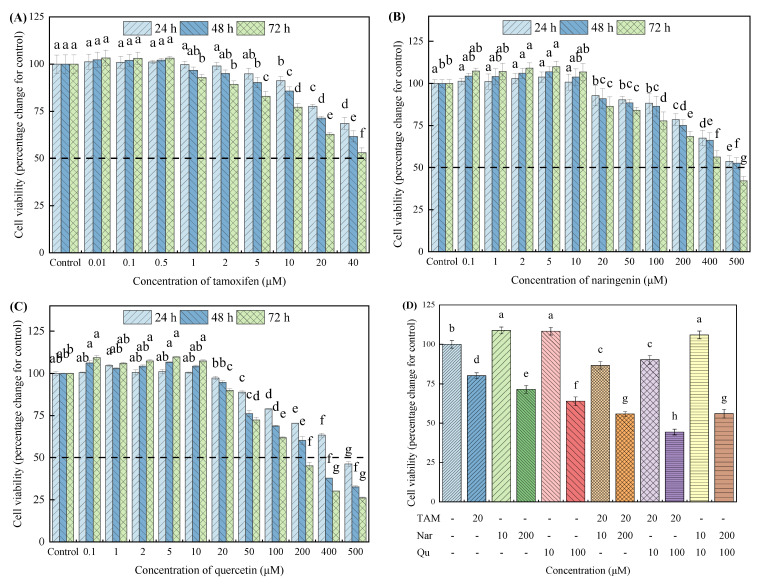
Effects of increasing concentrations of tamoxifen, naringenin and quercetin or their combined treatment on cell proliferation and cytotoxicity using CCK-8 assay in HepG2 cells. Cells were exposed to tamoxifen (**A**), naringenin (**B**) and quercetin (**C**) either alone or in combination (**D**) for 24, 48 and 72 h, respectively, and then the cell viability was measured using a CCK-8 assay according to the manufacturer instructions. Results are expressed as mean ± SD of three independent experiments (*n* = 6). ^a–h^: Different letters indicate significant differences (*p* < 0.05) in cell viability after cells were treated with tamoxifen, naringenin and quercetin alone or in combination. The hyphen (-) symbolizes HepG2 cells were not treated with tamoxifen, naringenin or quercetin. *TAM* means tamoxifen, *Nar* means naringenin, and *Qu* means quercetin, all of the abbreviations in this article are represented in this way.

**Figure 2 nutrients-14-05394-f002:**
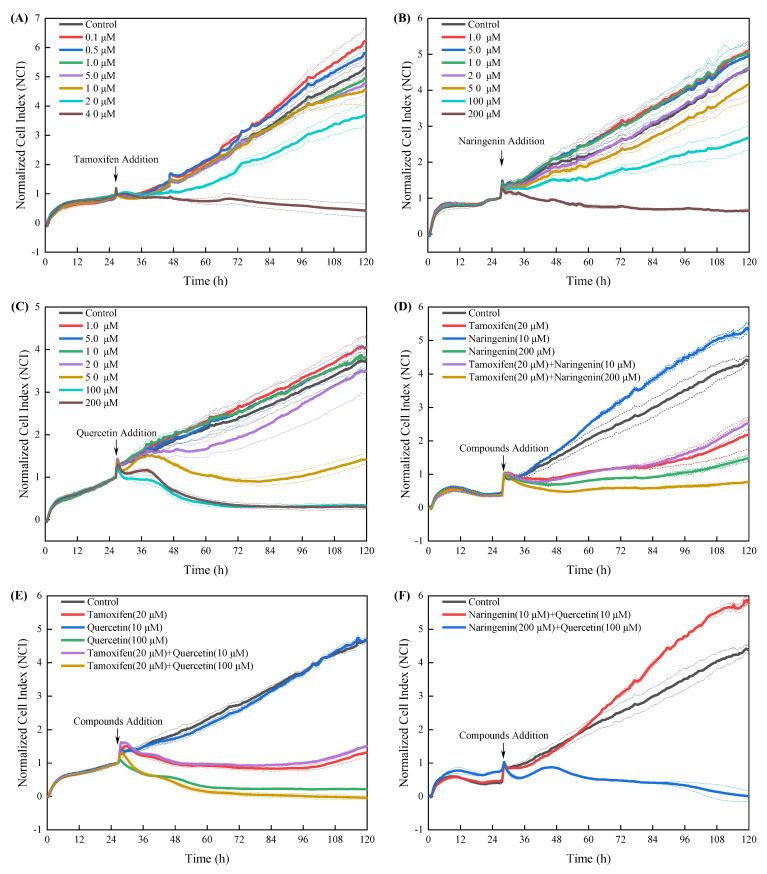
Effects of increasing concentrations of tamoxifen, naringenin and quercetin or their combined treatment on cell proliferation and cytotoxicity using RTCA assay. Cells were exposed to tamoxifen (**A**), naringenin (**B**) and quercetin (**C**) either alone or in combination (**D**–**F**) for up to 96 h, and the normalized cell index (NCI) was determined using real-time cellular impedance analysis monitoring equipment (xCELLigence RTCA S16) according to the manufacturer instructions. Results are expressed as mean ± SD of three independent experiments (*n* = 2).

**Figure 3 nutrients-14-05394-f003:**
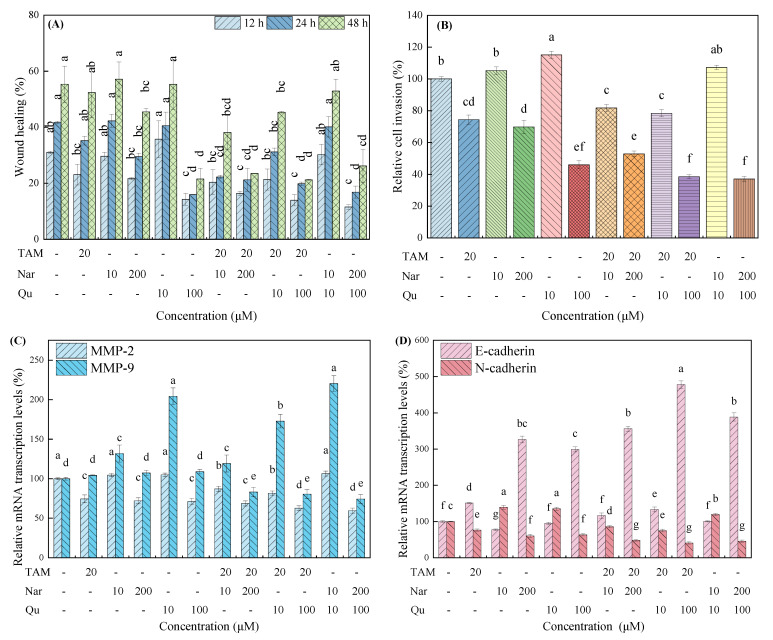
Effects of tamoxifen, naringenin and quercetin on cell migration and invasion in HepG2 cells. (**A**) Scratch healing following treatment with tamoxifen (20 μM), naringenin (10 and 200 μM) and quercetin (10 and 100 μM) alone or in combination for 12, 24 and 48 h. (**B**) Relative cell invasion ratio following treatment with tamoxifen, naringenin and quercetin for 24 h. (**C**,**D**) Tamoxifen, naringenin and quercetin regulated the gene expression of MMP-2, MMP-9, E-cadherin and N-cadherin in the mRNA transcription levels. Results are expressed as mean ± SD of three independent experiments (*n* = 3). ^a–g^: Different letters indicate significant differences (*p* < 0.05) in cell migration and invasion, as well as their related genes.

**Figure 4 nutrients-14-05394-f004:**
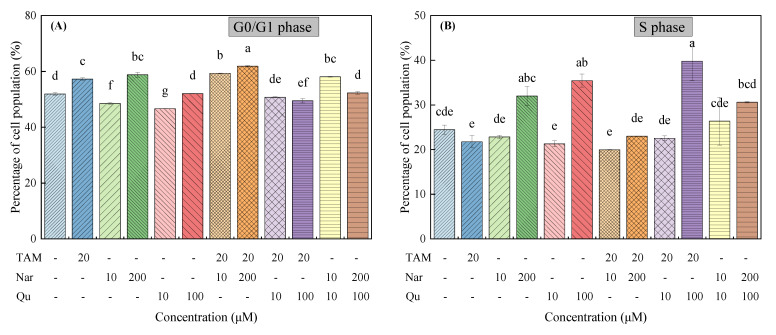
Tamoxifen, naringenin and quercetin regulated cell cycle progression in HepG2 cells. Cells were exposed to tamoxifen (20 μM), naringenin (10 and 200 μM) and quercetin (10 and 100 μM) either alone or in combination for 24 h. Percentage of cell population in the (**A**) G0/G1 phase, (**B**) S phase and (**C**) G2/M phase, respectively. (**D**) Tamoxifen, naringenin and quercetin regulated the mRNA transcription of cell cycle-related genes. Results are expressed as mean ± SD of three independent experiments (*n* = 3). ^a–h^: Different letters indicate significant differences (*p* < 0.05) in cell cycle distribution and their regulated genes.

**Figure 5 nutrients-14-05394-f005:**
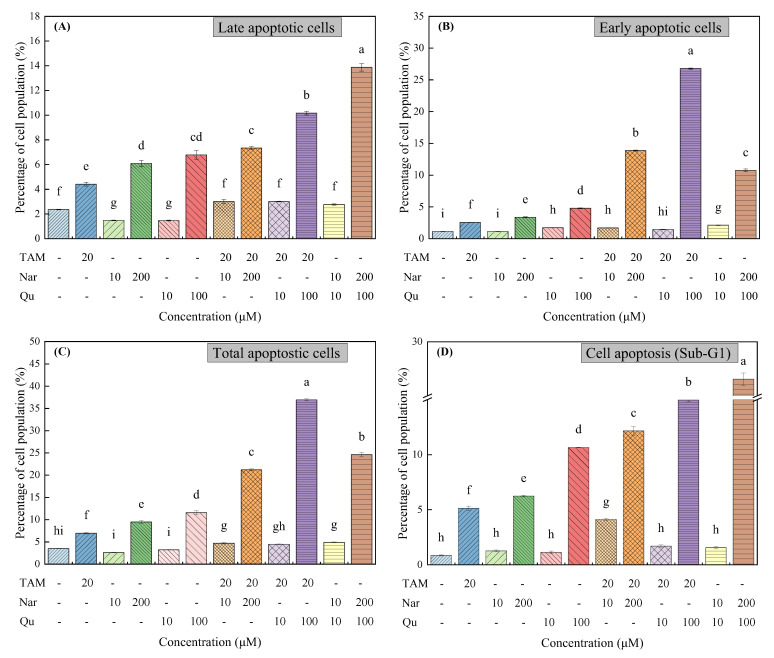
Tamoxifen, naringenin and quercetin regulated cell apoptosis in HepG2 cells. Cells were exposed to tamoxifen (20 μM), naringenin (10 and 200 μM) and quercetin (10 and 100 μM) either alone or in combination for 24 h. (**A**–**C**) Percentage of cell population in the different apoptotic stages. (**D**) Sub-G1 DNA content of cells analyzed with PI staining. Results are expressed as mean ± SD of three independent experiments (*n* = 3). ^a–i^: Different letters indicate significant differences (*p* < 0.05) in cell apoptosis.

**Figure 6 nutrients-14-05394-f006:**
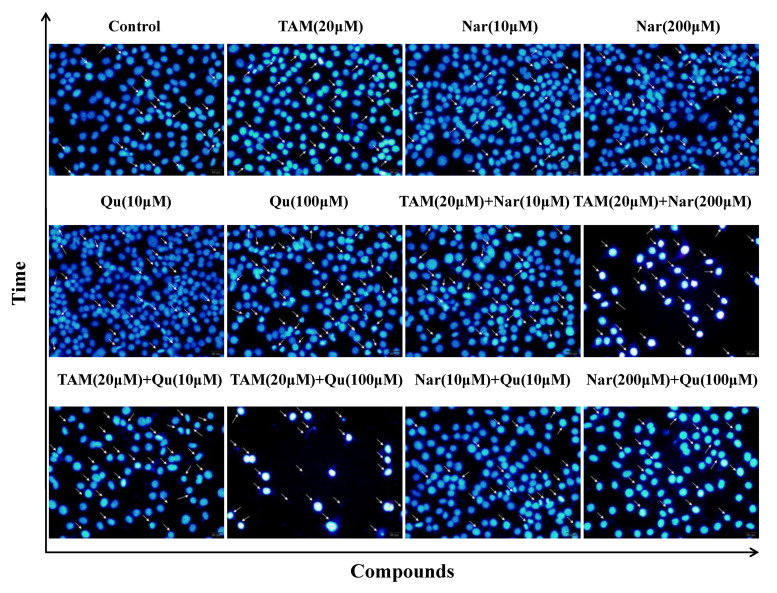
Representative fluorescence immunocytochemistry of HepG2 cells exposed to tamoxifen, naringenin and quercetin. Cells were stained with DAPI after treatment with tamoxifen (20 μM), naringenin (10 and 200 μM) and quercetin (10 and 100 μM) alone or in combination for 24 h. Then, cells were scrutinized for nuclear morphology at 400× magnifications, and the white arrows (→) indicate chromatin condensation and fragmentation in apoptotic cells.

**Figure 7 nutrients-14-05394-f007:**
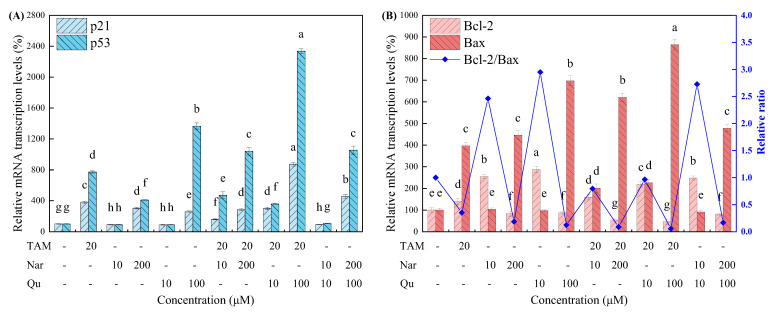
Tamoxifen, naringenin and quercetin regulated the mRNA transcription of the typical apoptosis-related genes in HepG2 cells. (**A**) The mRNA transcription levels of p21 and p53. (**B**) The mRNA transcription levels of Bcl-2 and Bax along with Bcl-2/Bax ratios. Results are expressed as mean ± SD of three independent experiments (*n* = 3). ^a–h^: Different letters indicate significant differences (*p* < 0.05) in cell apoptosis for 24 h.

**Figure 8 nutrients-14-05394-f008:**
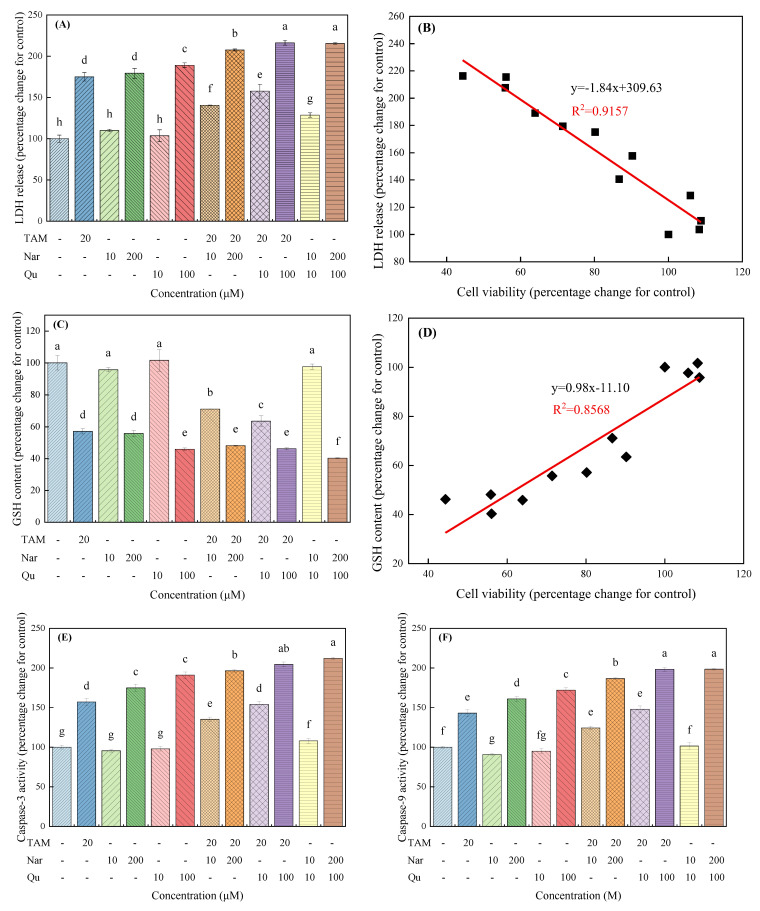
Effects of tamoxifen, naringenin and quercetin on LDH release, GSH content and caspase activity in HepG2 cells. Cells were treated with tamoxifen, naringenin and quercetin alone or in combination for 24 h. LDH release in the culture medium (**A**) and GSH content (**C**) were measured using an LDH cytotoxicity assay and glutathione reductase assay, respectively, according to the manufacturer instructions. The relative correlations between cell viability and LDH release (**B**) or GSH content (**D**) were depicted utilizing the linear fit. (**E**,**F**) The activities of caspase-3 and caspase-9 were measured using colorimetric assay kits. Results are expressed as mean ± SD of three independent experiments (*n* = 6). ^a–h^: Different letters indicate significant differences (*p* < 0.05) in LDH release, GSH content and caspase activity.

**Figure 9 nutrients-14-05394-f009:**
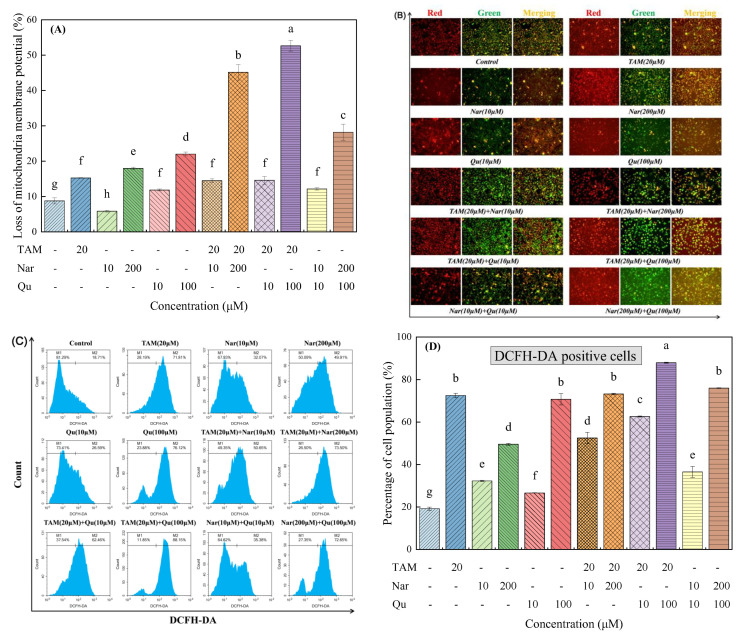
Tamoxifen, naringenin and quercetin regulated mitochondrial membrane potential and reactive oxygen species generation in HepG2 cells. Cells were exposed to tamoxifen (20 μM), naringenin (10 and 200 μM) and quercetin (10 and 100 μM) either alone or in combination for 24 h. (**A**) Percentage of mitochondrial membrane potential loss by JC-1 staining is represented as a bar diagram. (**B**) Loss of ΔΨm was monitored with JC-1 dye using fluorescence microscopy at a magnification of 200×. (**C**) Representative flow cytometry images showed that tamoxifen, naringenin and quercetin regulated ROS generation compared with the untreated control group. (**D**) The percentage of the cell population of DFCH-DA positive is represented as a bar diagram. Results are expressed as mean ± SD of three independent experiments (*n* = 3). ^a–h^: Different letters indicate significant differences (*p* < 0.05) in the loss of ΔΨm and ROS generation.

## Data Availability

The data are available upon request.

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
