# Peer review of "Naringenin and Quercetin Exert Contradictory Cytoprotective and Cytotoxic Effects on Tamoxifen-Induced Apoptosis in HepG2 Cells"

_nutrients, 2022, doi:10.3390/nu14245394_

Round 1

Reviewer 1 Report

Dear authors,

I have some comments that might help you improve details of your manuscript.

In many sections of the manuscript you refer to naringenin/quercetin as phytoestrogens, but this definition is given to molecules that are similar to estrogen, which to fit this description many isoflavones and naringenin derivatives could be considered as such, but not flavonols or flavanones (https://doi.org/10.1146/annurev.arplant.55.031903.141729), if you have any sources that consider naringenin or quercetin as phytoestrogen please specify this in your manuscript.

In figure 1A,B,C, please add the letters of the statistical analysis, even if you didn't observe significant differences. It will help your readers to understand and not assume things.

Also, please homogenize the letters of the statistical analysis in Figure 3.

Please place in a homogenized form the letters in Figure 4.

In the conclusions section, please remove the reference to Figure 8, I recommend this as conclusions should only focus on whether you proved or not your hypothesis.

Figure 5 seems too charged, I recommend you to section this figure in Figure 5 (Figures of percentage of cell population), Figure 6 (fluorescence immunochemistry), and Figure 7 (relative mRNA transcription levels figures).

Author Response

Dear Anonymous Reviewer,

On behalf of my co-authors, we would like to thank you a lot for allowing us to revise our manuscript "Naringenin and quercetin exert contradictory cytoprotective and cytotoxic effects on tamoxifen-induced apoptosis in HepG2 cells" (Manuscript No: nutrients-1913556). We appreciate your suggestions and constructive comments.

We have addressed the comments raised by you, and the amendments are highlighted in red in the revised manuscript. Point-by-point responses  can be found in the attachment.

All further suggestions and comments could be appreciated. We hope that the revised version of the manuscript is now acceptable for publication in Nutrients. We look forward to hearing from you soon.

Best regards,

Xiaoxia Yang, Ph.D. and Xuejun Pan, Ph. D.

Kunming University of Science and Technology

Email: [email protected]; [email protected]

Reviewer 2 Report

The purpose of this study was to determine the synergistic antitumor effect of tamoxifen and naringenin/quercetin in human liver carcinoma and to explore the possible molecular mechanisms involved, for which the HepG2 cell line was used as an in vitro model. The results obtained in the experiments carried out allowed the authors to conclude that there is a contradiction in the effects observed at high and low concentrations of naringenin and quercetin, because these phytoestrogens exerted cytoprotective and cytotoxic effects depending on their concentration and the effect of tamoxifen on liver cancer. human.

The article is quite interesting and complete, the stated objective is consistent with the experimental design and the tests carried out in vitro. Likewise, the results obtained clearly demonstrate that the effect of phytoestrogens on human liver cancer can be beneficial when acting in conjunction with tamoxifen, but under certain conditions.

Author Response

Response to Reviewer #2

Comments to the Authors (General comments)

  1. The purpose of this study was to determine the synergistic antitumor effect of tamoxifen and naringenin/quercetin in human liver carcinoma and to explore the possible molecular mechanisms involved, for which the HepG2 cell line was used as an in vitro model. The results obtained in the experiments carried out allowed the authors to conclude that there is a contradiction in the effects observed at high and low concentrations of naringenin and quercetin, because these phytoestrogens exerted cytoprotective and cytotoxic effects depending on their concentration and the effect of tamoxifen on human liver cancer.

Answer: We would like to express our sincere gratitude for your contribution to our manuscript. As you said, the present study shows for the first time that low concentrations of naringenin and quercetin can stimulate the proliferation of HepG2 cells and then inhibit their proliferation. Moreover, naringenin and quercetin exhibit contradictory cytoprotective and cytotoxic effects on tamoxifen-induced antiproliferation in human hepatic cancer HepG2 cells.

  1. The article is quite interesting and complete, the stated objective is consistent with the experimental design and the tests carried out in vitro. Likewise, the results obtained clearly demonstrate that the effect of phytoestrogens on human liver cancer can be beneficial when acting in conjunction with tamoxifen, but under certain conditions.

Answer: Thanks very much for your careful review and high recognition of our manuscript.